# Nicotine rebalances NAD+ homeostasis and improves aging-related symptoms in male mice by enhancing NAMPT activity

Liang Yang [1,2,3], Junfeng Shen[1,4], Chunhua Liu[1,2,3], Zhonghua Kuang[1,5,6], Yong Tang[1,2], Zhengjiang Qian [1,2], Min Guan[1,7], Yongfeng Yang[1,5,6], Yang Zhan [1,2], Nan Li[1,4,8] & Xiang Li [1,3] ✉

Imbalances in NAD+ homeostasis have been linked to aging and various diseases. Nicotine, a metabolite of the NAD+ metabolic pathway, has been found to possess anti-inflammatory and neuroprotective properties, yet the underlying molecular mechanisms remained unknown. Here we find that, independent of nicotinic acetylcholine receptors, low-dose nicotine can restore the age-related decline of NAMPT activity through SIRT1 binding and subsequent deacetylation of NAMPT, thus increasing NAD+ synthesis. $^{18}$F-FDG PET imaging revealed that nicotine is also capable of efficiently inhibiting glucose hypermetabolism in aging male mice. Additionally, nicotine ameliorated cellular energy metabolism disorders and deferred age-related deterioration and cognitive decline by stimulating neurogenesis, inhibiting neuroinflammation, and protecting organs from oxidative stress and telomere shortening. Collectively, these findings provide evidence for a mechanism by which low-dose nicotine can activate NAD+ salvage pathways and improve age-related symptoms.

Nicotinamide adenine dinucleotide (NAD) homeostasis is organized and coordinated in many physiological statuses. However, with advancing age, the equilibrium of NAD+ homeostasis is impaired, leading to a considerable decline, correlating with age-related defects[1–5]. To counteract this, two strategies have been proposed: reducing the activity of NAD+-consuming enzymes[6,7] and increasing the biosynthesis of NAD+ by supplying NAD+ precursors and activating NAD+ synthetic enzymes.

Previous studies showed that supplementing NAD+ precursors, such as nicotinamide mononucleotide (β-NMN), nicotinamide riboside (NR), nicotinamide (NAM), or Niacin (NA), were efficient for anti-aging[8–10]. In mammalian cells, most NAD+ precursors were "reclaimed" to NAD+ via the rate-limiting enzyme nicotinamide phosphoribosyl

transferase (NAMPT) of the salvage pathway. As NAMPT activity decreases with age, it results in NAD+ depletion and cognitive impairment[11]. Consequently, the NAMPT enhancers can effectively increase NAD+ levels and prevent neuronal degeneration[12–14]. Moreover, supplementing the high active extracellular NAMPT (eNAMPT) also increases NAD+ biogenesis and aging-related symptoms[15–17]. Therefore, the enhancement of NAMPT activity could be a potential strategy for preventing aging and age-associated diseases arising from NAD+ decline[11,18,19].

Interestingly, nicotine is a secondary metabolite of the NAD+ biosynthesis and the NAD+ pathway is also coordinately regulated with nicotine biosynthesis[20,21]. Furthermore, the NAD+ salvage pathway is conserved across a variety of species, including bacteria, plants, yeasts,

[1]Shenzhen Institute of Advanced Technology, Chinese Academy of Sciences, Shenzhen 518055, China. [2]Brain Cognition and Brain Disease Institute (BCBDI), Shenzhen, China. [3]Guangdong Provincial Key Laboratory of Brain Connectome, Shenzhen Key Laboratory of Viral Vectors for Biomedicine, Shenzhen, China. [4]Shenzhen Institute of Synthetic Biology, Shenzhen, China. [5]Institute of Biomedical and Health Engineering, Shenzhen, China. [6]Paul C. Lauterbur Research Center for Biomedical Imaging, Shenzhen, China. [7]Institute of Biomedicine and Biotechnology, Shenzhen, China. [8]Chinese Academy of Sciences (CAS) Key Laboratory for Quantitative Engineering Biology, Shenzhen, China. ✉e-mail: xiang.li@siat.ac.cn

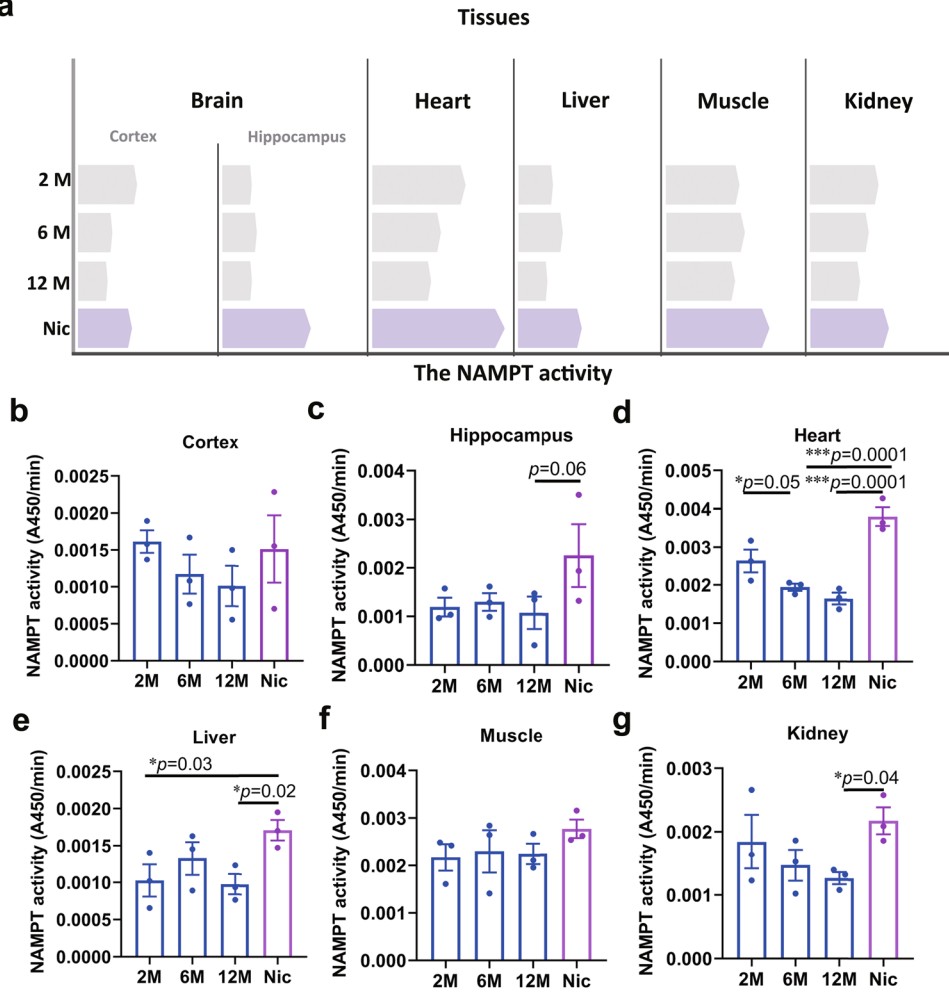

**Fig. 1 | Nicotine enhanced NAMPT activity in aged mice. a** Schematic diagram of the changes in NAMPT activity and nicotine enhancing NAMPT activity in various tissues (the bars represent the mean value of NAMPT activity). Statistical diagram of NAMPT activity change of different tissues: **b** Cortex, **c** Hippocampus, **d** Heart, **e** Liver, **f** Muscle, **g** Kidney. Data are means ± SEM ($n = 3$, biologically independent samples/group); $p$ values were determined by one-way ANOVA with Tukey's multiple comparisons test (**b–g**).

and mammals[20]. Consequently, we hypothesized that nicotine may play a role in the NAD$^+$ salvage pathway. It has been discovered that minuscule amounts of nicotine, as an activator of NAD$^+$ biogenesis, can markedly improve NAMPT activity and NAD$^+$ synthesis, leading to improved glucose metabolism and cognitive function, as well as aging symptoms, in male mice.

## Long-term, nanogram-trace of nicotine boosts NAMPT activity and enhanced the SIRT1 deacetylation of NAMPT in aged mice

To examine the effect of nicotine on NAMPT activity in different tissues of aging mice, we tested NAMPT activity in the hippocampus, cortex, heart, liver, muscle, and kidney at 2, 6, and 12 months of age. We found that the activity of NAMPT in mouse tissues gradually decreased with increasing age. After nicotine administration in drinking water (2 μg/mL) from 6 to 12 months, nicotine significantly restored the NAMPT activity in all the tissues (Fig. 1a–g). Consistently, we detected β-NMN levels in brain and liver tissues by LC-MS and found that β-NMN levels were significantly increased after nicotine administration (Supplementary Fig. 9c).

It was previously known that NAMPT activity is affected by its acetylation level. To examine the acetylation levels of NAMPT in tissues of mice of different ages, we precipitated acetylated-NAMPT from various tissues and found that between 2 to 6 months, the acetylation

levels of NAMPT have not changed in most of tissues of mice, except kidneys in which the hyperacetylation of NAMPT elevated significantly as early as 6 months of age (Fig. 2k). At 12 months, acetylation levels of NAMPT reached high levels in all tissues. Notability, nicotine could significantly reduce the acetylation level of NAMPT, and the deacetylation effect of nicotine on NAMPT was the most significant in heart (Fig. 2a–k).

Furthermore, previous studies showed that NAMPT activity could be regulated by SIRT1 deacetylation[15]. We then tested the SIRT1 interacted with NAMPT from 2 to 12 months, and found that the binding of SIRT1-NAMPT in all tissues at 2 and 6 month were much higher than 12 months, which was consistent with the results of changes in NAMPT acetylation levels in different tissues at different months. Interestingly, nicotine also promoted the binding of SIRT1 to NAMPT in all of the tested tissues (Fig. 3a–k). In summary, these results reveal a surprisingly direct role for nicotine in regulating NAD$^+$ salvage pathway, demonstrating that nicotine promotes the binding of SIRT1 to NAMPT and enhances the deacetylation of NAMPT and finally enhances NAMPT activity.

## Nicotine enhanced SIRT1 binding of NAMPT in a dose-dependent manner in the aged cells

To further validate the binding and biological function of SIRT1 to NAMPT via nicotine, we calculated the exact concentration of nicotine

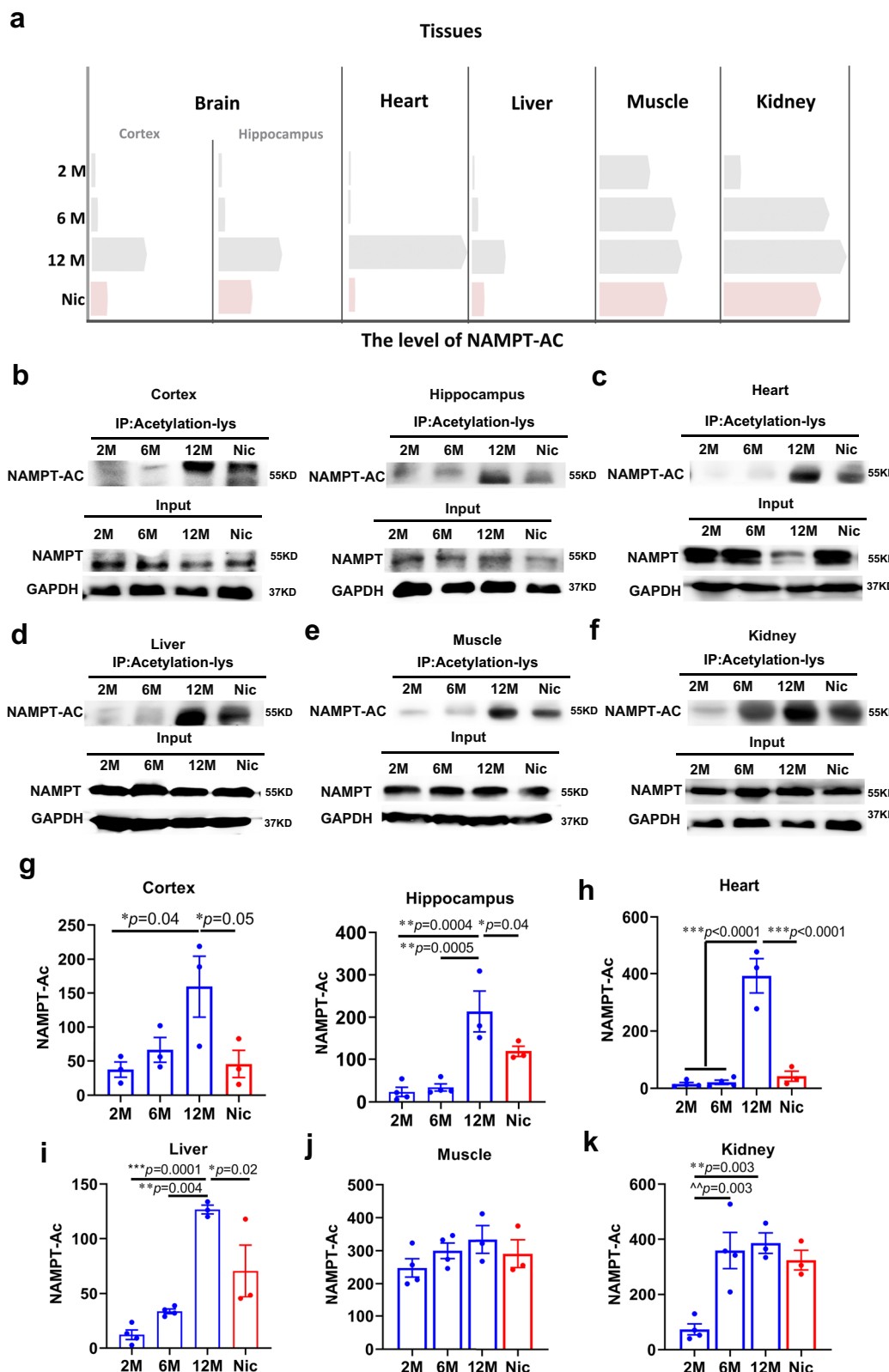

**Fig. 2 | Nicotine increased NAMPT deacetylation in aged mice. a** Schematic diagram of the changes in NAMPT acetylation levels and nicotine decreased NAMPT acetylation in various tissues (the bars represent the mean value of NAMPT acetylation). The acetylation levels of NAMPT in different tissues: **b** Cortex and Hippocampus, **c** Heart, **d** Liver, **e** Muscle, **f** Kidney. Statistical diagram of NAMPT acetylation levels in different tissues: **g** is a quantification of **b** (Cortex and Hippocampus), **h** is a quantification of **c** (Heart), **i** is a quantification of **d** (Liver), **j** is a quantification of **e** (Muscle), **k** is a quantification of **f** (Kidney). Data are means ± SEM ($n = 3–4$, biologically independent samples/group); $p$ values were determined by one-way ANOVA with Tukey's multiple comparisons test (**g**–**k**).

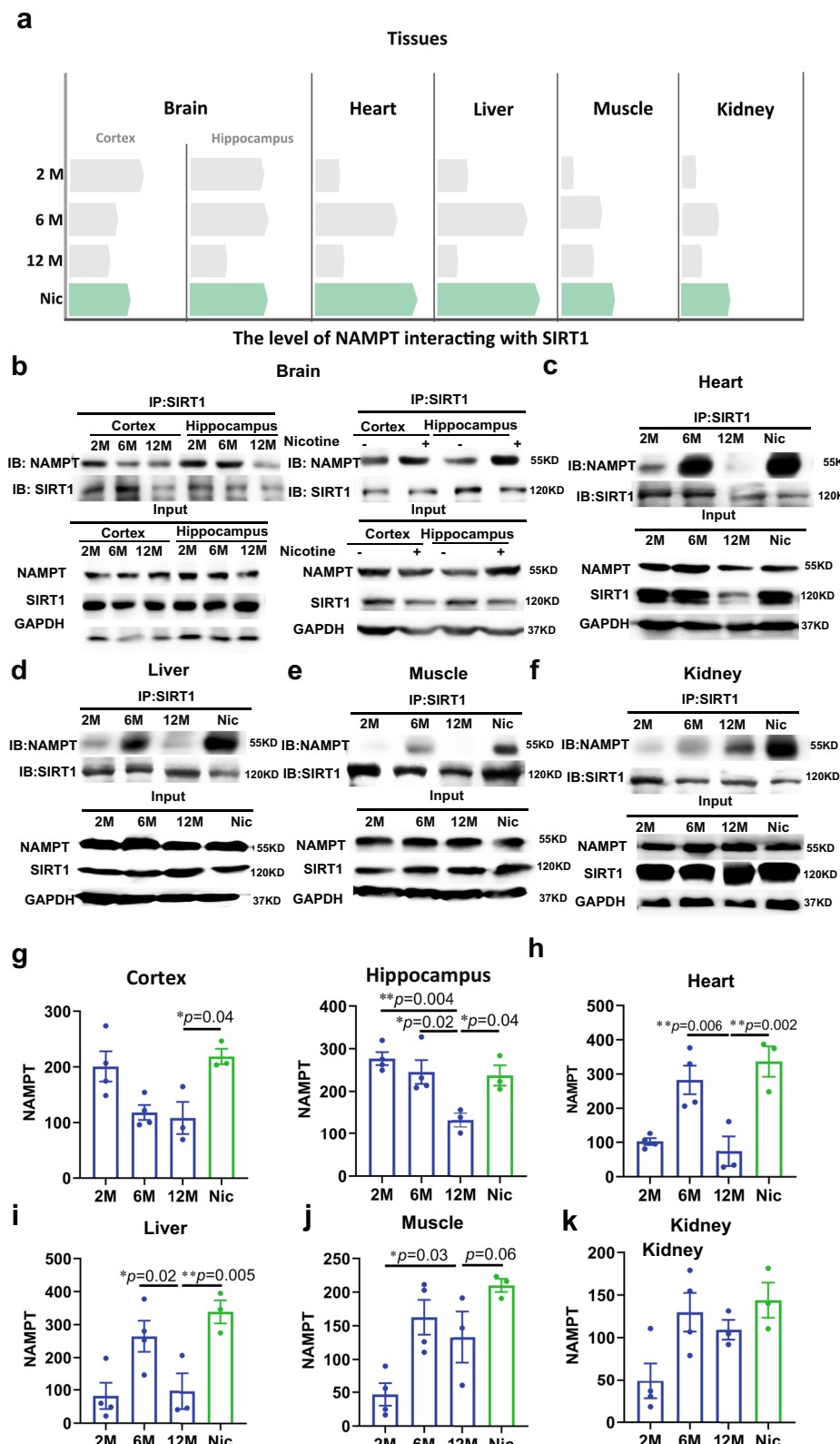

**Fig. 3 | Nicotine promoted SIRT1 binding with NAMPT in aged mice. a** Schematic diagram of the levels of NAMPT interaction with SIRT1 in various tissues (the bars represent the mean value of NAMPT-SIRT1 binding levels). The binding of SIRT1 and NAMPT in different tissues: **b** Cortex and Hippocampus, **c** Heart, **d** Liver, **e** Muscle, **f** Kidney. Statistical diagram of NAMPT acetylation levels of different tissues: **g** is a quantification of **b** (Cortex and Hippocampus), **h** is a quantification of **c** (Heart), **i** is a quantification of **d** (Liver), **j** is a quantification of **e** (Muscle), **k** is a quantification of **f** (Kidney). Data are means ± SEM ($n$ = 3–4, biologically independent samples/group); $p$ values were determined by one-way ANOVA with Tukey's multiple comparisons test (**g**–**k**).

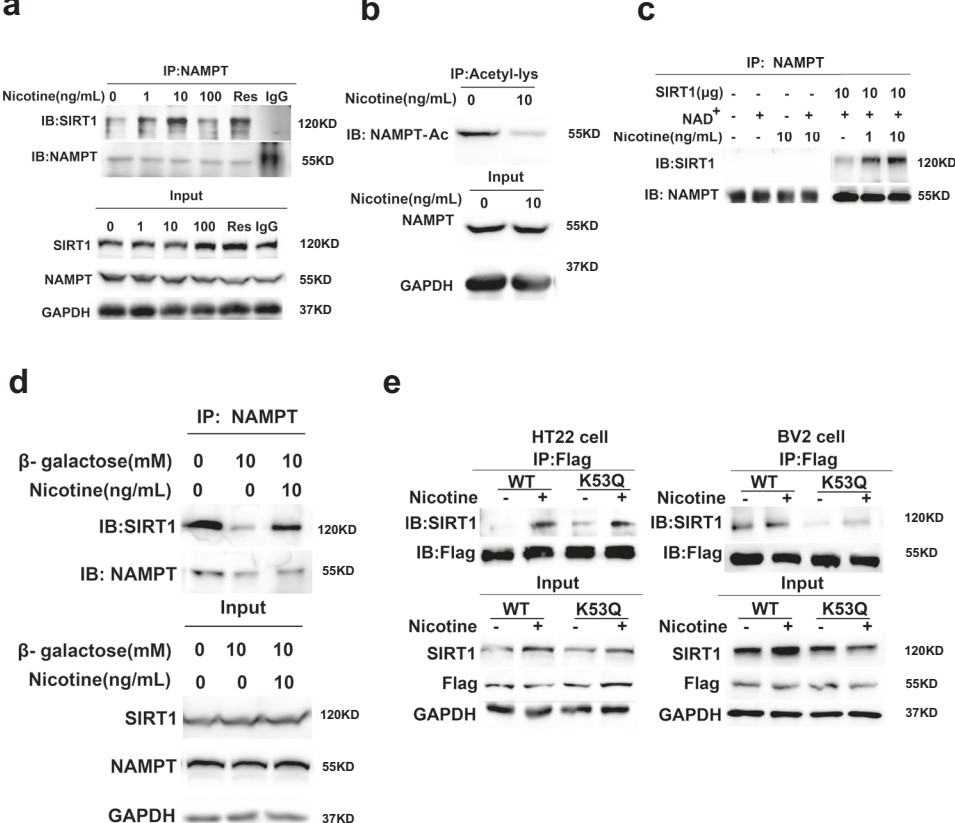

**Fig. 4 | Nicotine increased SIRT1 interacts with NAMPT and deacetylation of NAMPT. a** The enhancement of SIRT1 and NAMPT interaction by nicotine and resveratrol (100 nmol/L). IgG antibodies were used as a negative control. **b** The acetylation levels of NAMPT in HT22 cells with or without nicotine for 48 h. **c** Co-immunoprecipitation of purified protein SIRT1 with cellular NAMPT after nicotine treatment in vitro and analyzed by western blot. **d** Nicotine recovers β-galactose-impaired SIRT1-NAMPT interaction in HT22 cells. **e** Nicotine enhances SIRT1-NAMPT interaction in NAMPT-K53Q mutation cells. All western blotting images and graphs are representative of three independent experiments.

in the tissues during this chronic administration. We directly measured the nicotine content of the aged tissue using LC-MS. The nicotine content in the liver and brain were about 15 ng/g, and about 0.25 ng/g in the serum (Supplementary Fig. 1a). Based on the in vivo concentrations, we investigated the effect of nicotine on NAD$^+$ regulation in vitro by using an appropriate range of concentrations of 10–20 ng/mL of nicotine in cultured cells.

Using the neuronal cell line HT22 as a model system, we asked whether nicotine could affect SIRT1 binding of NAMPT and contributed to the NAD$^+$ synthesis pathway. We found that low-dose of nicotine (1 and 10 ng/mL) significantly enhanced SIRT1 binding to NAMPT; in contrast, a high dose of nicotine (100 ng/mL) did not change this interaction (Fig. 4a). We also found that nicotine did not affect NAMPT expression in transcription and translation level (Supplementary Fig. 1b–d). Using the acetylation-lysine antibody to immunoprecipitate acetylated NAMPT; we found a significantly decreased acetylation level of NAMPT (Fig. 4b). Consistently, the immunofluorescence showed that nicotine decreased the nuclear acetylation of HT22 cells (Supplementary Fig. 2a–c) and this reduction of acetylation might be due to the cytoplasmic SIRT1 translocation to the nucleus upon nicotine supplement (Supplementary Fig. 2d, f).

To further verify the modulating effect of nicotine, we precipitated NAMPT proteins from HT22 cells and incubated them with the purified SIRT1 proteins in the presence or absence of nicotine, and found that nicotine enhanced endogenous NAMPT binding with SIRT1 in a dose-dependent manner (Fig. 4c). Similar to in vivo studies, 10 ng/mL of nicotine increased NAMPT activities (Supplementary Fig. 3a). We also measured the content of β-NMN with LC-MS and found that nicotine

increased β-NMN levels in a dose-dependent manner (Supplementary Fig. 3b). The results also showed that 1 ng/mL of nicotine slightly enhanced NAD$^+$ levels, whereas 10 ng/mL significantly increased NAD$^+$ levels. Surprisingly, 100 ng/mL of nicotine had no significant effect, and 500 ng/mL of nicotine inhibited NAD$^+$ levels in the cells (Supplementary Fig. 3c). Consistent with NAD$^+$ elevation, the NAD$^+$-NADH redox fluorescent biosensor in the transfected cells revealed that nicotine significantly increased the ratio of NAD$^+$/NADH in both the cytoplasm and the nucleus (Supplementary Fig. 3d). Moreover, 10 ng/mL of nicotine rescued the SIRT1-NAMPT interaction in β-galactose-induced aged cells (Fig. 4d) and prevented β-galactose-induced NAD$^+$ decline in the senescent cells. To further explore the regulation of nicotine on hyperacetylation of NAMPT, we constructed the NAMPT-K53Q cell to simulate hyperacetylation of NAMPT during aging. SIRT1 bound to NAMPT-K53Q was significantly less than the wild type in both HT22 and BV2 cells. However, nicotine could partially restore the binding of SIRT1 to NAMPT (Fig. 4e). These data indicated that the low-dose of nicotine enhanced endogenous NAMPT binding with SIRT1 in a dose-dependent manner and boosted the β-NMN and NAD$^+$ production in aged cells.

## Nicotine specifically regulates the SIRT1-NAMPT interaction

The Sirtuins are the main consumer of NAD$^+$ and their activity was limited by NAD$^+$ levels. In consistent with NAD$^+$ level increase by nicotine treatment, we also found that nicotine significantly increased the brain sirtuins (SIRT1, SIRT6, and SIRT7) activities (Fig. 5a), while the expression of these sirtuins was not changed (Supplementary Fig. 4a). Therefore, we tested whether nicotine could specifically induce SIRT1

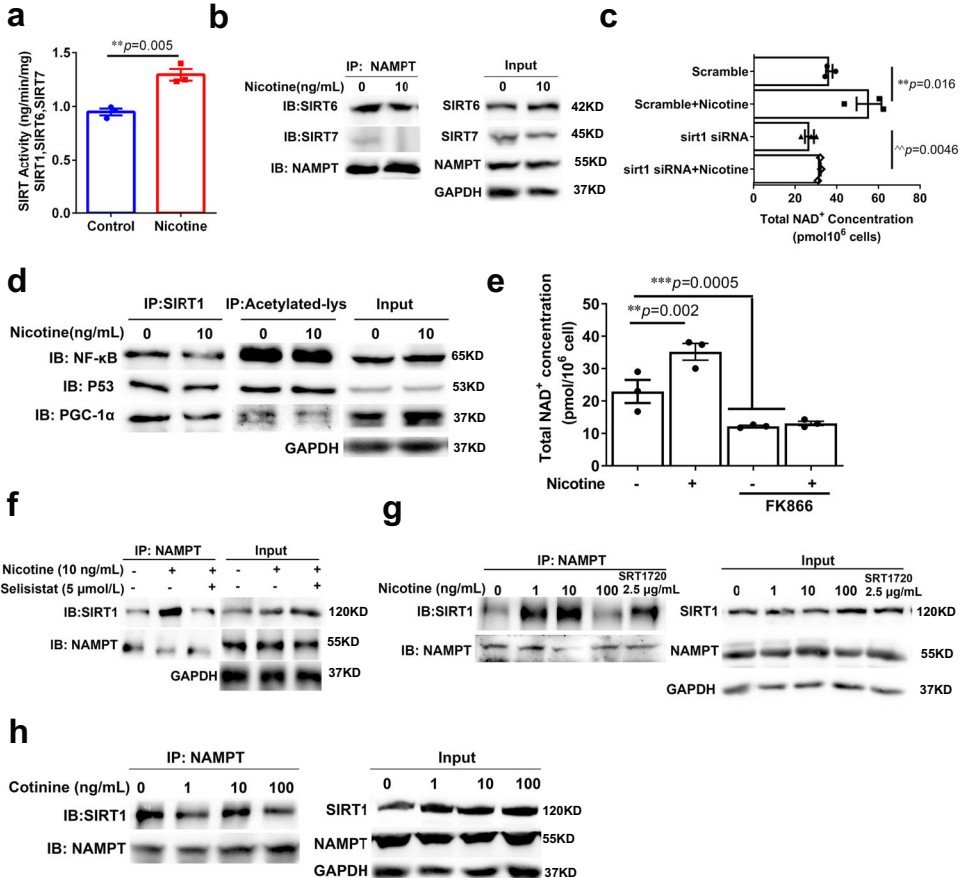

**Fig. 5 | The specific effect of nicotine on SIRT1-NAMPT interaction. a** Nuclear SIRTUIN (SIRT1, SIRT6, SIRT7) activity of aged mice brain with or without nicotine at 13 months of age (n = 3, biologically independent samples/group). **b** Nicotine has no district effects on NAMPT-SIRT6/ SIRT7 interaction. **c** The total NAD⁺ levels of SIRT1 knockdown cells after nicotine treatment (n = 3, biologically independent samples/group). **d** Nicotine (10 ng/mL) has no district enhancement on SIRT1 interact with other substrates (p53, PGC-1, NF-κB). **e** The NAMPT inhibitor FK866 abolished the effect of nicotine (n = 3, biologically independent samples/group). **f** SIRT1 inhibitor has no district effect on SIRT1-NAMPT interaction. **g** The effect of SIRT1 activator SRT1720 (2.5 μg/mL) and inhibitor Selisistat (5 μmol/L) on SIRT1-NAMPT interaction. **h** Cotinine has no effect on SIRT1-NAMPT interaction. All western blotting images and graphs are representative of three independent experiments. Data are means ± SEM. p values were determined by two-sided Student's t test (**a**), or one-way ANOVA with Tukey's multiple comparisons tests (**c**, **e**).

binding of NAMPT, but not the other sirtuins members. SIRT6 and SIRT7 were precipitated with NAMPT in HT22 cells. Notably, nicotine did not increase the interaction of NAMPT with SIRT6 or SIRT7 (Fig. 5b).

To examine whether SIRT1 as the predominant effector of NAD⁺ generation was specifically affected by nicotine, we knocked down SIRT1 in HT22 cells and found that NAD⁺ levels were significantly dropped, while nicotine could not rescue this reduction (Fig. 5c and Supplementary Fig. 4b). Previous studies showed that NAMPT could also be a substrate of SIRT6[22]. As a control, we knocked down SIRT6 and found that the NAD⁺ level was still increased in the presence of nicotine (Supplementary Fig. 4c, d).

SIRT1 deacetylates numerous substrates, including NF-κB, p53, and PGC-1α[23–25]. To explore whether SIRT1 specifically deacetylates NAMPT rather than other substrates under nicotine treatment, we performed the precipitation of NF-κB, p53, and PGC-1α with SIRT1 and acetylation-lysine, and found that nicotine (10 ng/mL) did not increase the SIRT1 binding or the deacetylation level of NF-κB, p53, and PGC-1α (Fig. 5d). Therefore, nicotine specifically mediates SIRT1-NAMPT interaction to drive NAD⁺ synthesis.

To investigate whether SIRT1 or NAMPT bioactivity was involved in the effect of nicotine on SIRT1-NAMPT interaction, we incubated HT22 cells with NAMPT or SIRT1 inhibitors respectively. We found that, although the inhibitors of NAMPT dramatically suppressed the elevation of NAD⁺ by nicotine (Fig. 5e), FK866 or Nampt-IN-1 did not change nicotine's stimulatory effect on SIRT1-NAMPT interaction

(Supplementary Fig. 4e). On the other hand, SIRT1 inhibitor, Selisistat, inhibited SIRT1 binding to NAMPT and abolished the effect of nicotine (Fig. 5f). These results suggest that bioactive SIRT1 is required for nicotine-induced NAD⁺ enhancement. SRT1720 and resveratrol are well-known activators of SIRT1. We compared the effect of nicotine with these two SIRT1 activators and found that SRT1720 and resveratrol were required much higher concentrations for promoting SIRT1-NAMPT interaction than nicotine (Fig. 5g, h). In contrast, nicotine neither directly inhibits/activates SIRT1 nor affects SIRT1 expression (Supplementary Fig. 4f, g). Hence, nicotine might solely enhance SIRT1 interaction with NAMPT to boost NAD⁺ synthesis, and the increased NAD⁺ in turn enhances SIRT1 activity and regulates NAMPT activity in a positive feedback manner.

Cotinine is the main metabolic product of nicotine[26]. To investigate whether cotinine could regulate the SIRT1-NAMPT interaction, we directly measured the content of cotinine was about 14.7 ng/g in the aged brain with LC-MS (Supplementary Fig. 4h). Based on this concentration, the HT22 cells were treated with cotinine at 1, 10, 100 ng/mL, and we found that cotinine did not affect the SIRT1-NAMPT interaction (Fig. 5g).

## The effect of nanogram-trace of nicotine was independent of brain nAChR activation

Generally, the biological effects of nicotine were considered to go through the nicotinic acetylcholine receptor (nAChRs)[27–34]. However,

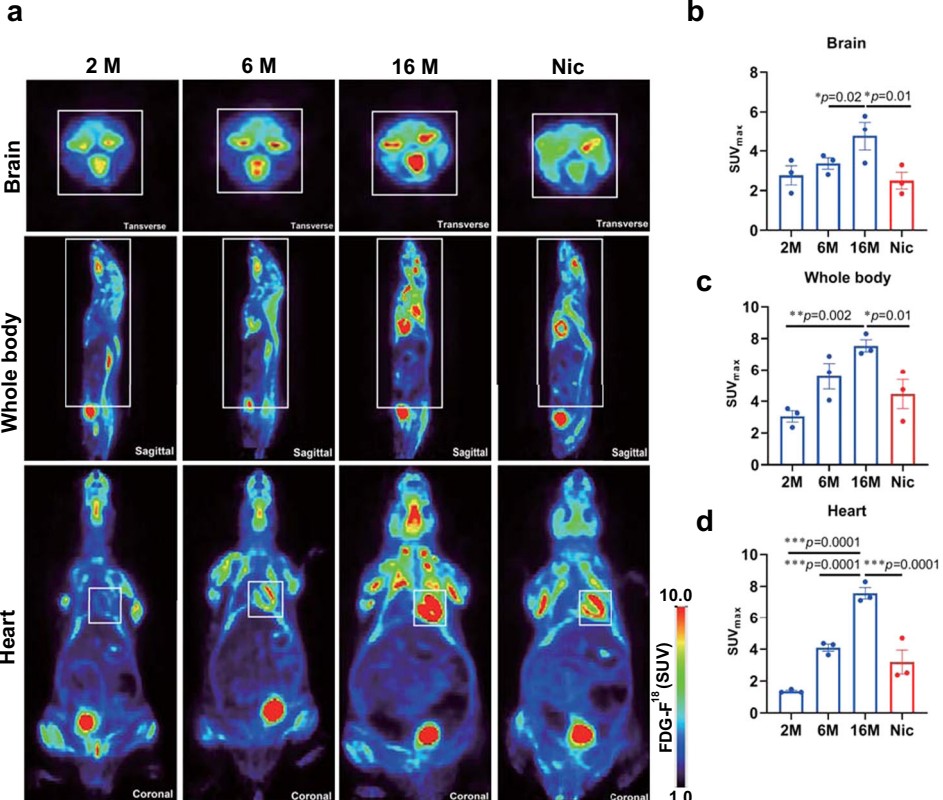

**Fig. 6 | Nicotine ameliorates aging-induced systemic glucose hypermetabolism. a** ROI-wise regional of $^{18}$F-FDG uptake in 2, 6, and 16 months age C57BL/6 mice compared with 16 months age mice with nicotine administration (the white box marks the ROI. The ROI: Transverse: Brain; Sagittal: whole body, Coronal: Heart). **b–d** Plots show the tissues SUV of $^{18}$F-FDG. Data are means ± SEM ($n$ = 3, biologically independent samples/group). $p$ values were determined by one ANOVA with Tukey post hoc correction (**b–d**).

the concentrations of nicotine utilized in this study were much lower than EC50 of nAChRs (0.5–100 μM) (Supplementary Fig. 1a). To rule out the possibility that the dose of nicotine in our study could change expression or activity of the nAChRs, we examined the transcription levels of nicotinic cholinergic receptors. The gene expression of nicotinic cholinergic receptor α and β subunits were not altered at all (Supplementary Fig. 1d). In addition, since the nAChR activation permeabilizes $Ca^{2+}$ [35], we transfected a fluorescence $Ca^{2+}$ indicator (GCamP6s) in HT22 cells to monitor the $Ca^{2+}$ permeability after nicotine treatment, which were subsequently treated with nicotine at 1, 10, 100 ng/mL, and 1 mg/mL. Compared with 1 mg/mL nicotine, 1–100 ng/mL nicotine did not increase $Ca^{2+}$ entry (Supplementary Fig. 1e, f). D-Tubocurarine chloride pentahydrate has been used to inhibit nAChRs activity, confirming that nicotine can increase the $NAD^+$ levels independent of nAChRs activation. By immunoprecipitation assay, it was found that the inhibition of nAChRs activity did not affect the nicotine promoting the binding effect of SIRT1 and NAMPT (Supplementary Fig. 1g). Therefore, the beneficial effects of nanogram-trace nicotine were independent of nAChRs activation.

## Nicotine can reverse the glucose hypermetabolism of the ageing mice

Senescence cells promote chronic inflammation through senescence-related secretory phenotype (SASP), leading to the decline of $NAD^+$ and NMN levels. The inflammatory response of senescent cells was closely related to hypermetabolism[36]. We used PET imaging glucose analog $^{18}$F-FDG to detect the glucose metabolism of the aged mice[37]. All groups were selected after completion of the VOI-based analysis. Consistent with previous studies, we found that glucose hypermetabolism in the brain, heart, and whole body gradually increased with

aging, and the $^{18}$F-FDG intake of 16-month-old mice was significantly higher than that of 2-and 6-month-old mice (Fig. 6a–d). However, we found that nicotine could dramatically decrease the high glucose metabolism status caused by aging in the brain, heart, and body. After administration of nicotine, the $^{18}$F-FDG uptake levels of old mice could return to the near 6-month-old level (Fig. 6a–d). These data indicate that nicotine could significantly reverse glucose hypermetabolism in various aging tissues, and remodel the types and levels of energy metabolism pathways by restoring the of $NAD^+$ level in aging tissues.

## Nicotine can improve energy metabolism by regulating NAMPT activity in aged cells

To determine how nicotine remodeled energy metabolism of different ageing cells, we transfected NAMPT-K53Q plasmids into HT22 and BV2 microglia to simulate the senescence of cells with different metabolic states, and then we carried out the extracellular flux assays using the Seahorse analyzer to dynamically track the metabolic changes in NAMPT-K53Q mutation cells treated with nicotine[15]. The glycolytic and mitochondrial respiration flux assays were measured by using Glycolysis stress test and Cell Mito stress test. We found that nicotine had no significant effect on aerobic respiration and glycolysis in original BV2 (Supplementary Fig. 5a, c) and HT22 cells (Supplementary Fig. 5b, d). Compared with the NAMPT-Flag group, the NAMPT-K53Q significantly inhibited Maximal respiration and Spare Respiratory Capacity (Fig. 7a) and elevated the Glycolysis and Glycolytic capacity function (Fig. 7b) in BV2 cells. In HT22 cells, NAMPT-K53Q significantly reduced Maximal Respiration, Spare Respiratory Capacity and Proton leak and elevated ATP production, but it does not affect glycolysis functions (Fig. 7c, d). Notably, we observed that nicotine could reverse the dysfunction of glycolytic and mitochondrial respiration in BV2 cells (Fig. 7a, b) and

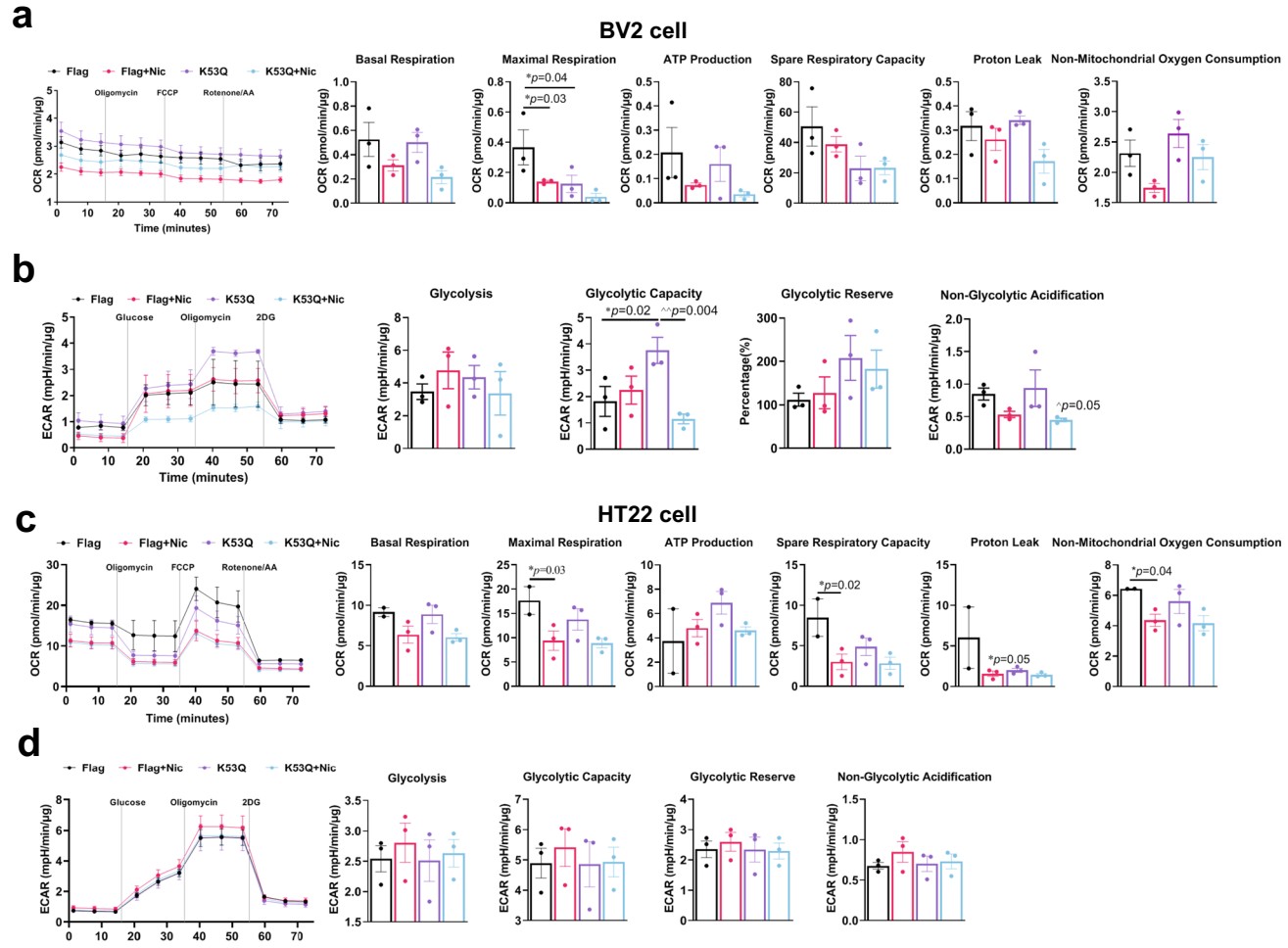

**Fig. 7 | Nicotine improved the OCR and ECAR in NAMPT-K53Q mutation cells.** The effects of nicotine on OCR measurement **a** and ECAR measurement **b** in BV2 cells ($n = 3$, biologically independent samples/group). The effects of nicotine on OCR measurement **c** and ECAR measurement **d** in HT22 cells ($n = 2$–3, biologically independent samples/group). The quantitative analysis of OCR including, basal respiration, ATP-linked respiration, proton leak, maximal respiration, spare respiratory capacity, non-mitochondrial oxygen consumption. The quantitative analysis of ECAR including, glycolysis, glycolytic capacity, glycolytic reserve and non-glycolytic acidification. Data are means ± SEM. $p$ values were determined by one-way ANOVA with Tukey's multiple comparisons tests (**a**–**d**).

slightly improve mitochondrial respiration in HT22 (Fig. 7c). We next examined the effects of nicotine on cell migration and found that the NAMPT-K53Q mutations inhibited the migration of HT22 cells and increased in BV2 cells, while nicotine could improve the migration in both Flag and NAMPT-K53Q mutations cells (Supplementary Fig. 6a, b), but has no effect on HT22 cells migration (Supplementary Fig. 6b). These data suggest that energy metabolism dysfunction is caused by hyperacetylation of NAMPT, and that low-dose nicotine may improve cell energy metabolism and associated physiological activity.

## Nicotine improved neurogenesis and inhibited inflammation in the aged mice

NAD[+] is required for over 500 enzymatic reactions and plays key roles in the regulation of almost all major biological processes[38]. To further explore the beneficial effect of nicotine on biological processes in the aged brain, we performed genome-wide RNA sequencing (RNA-seq) in the brain from 13 months old mice treated with nicotine for 6 months. By "spearman rank correlation" and principal component analyses (PCA) (Supplementary Fig. 7a, b), there are 222 upregulated genes and 51 downregulated genes between the control group and nicotine group ($p$ value <0.05) (Fig. 8a). The Gene KEGG pathways related to "Signal transduction", "Aging", "Immune system", "Transcription" were

the most significantly changed in genome-wide transcriptional profiles ($p$ value <0.05) (Fig. 8b).

We next performed key-driven analysis and the 110 DEGs ($p$ value <0.01) and narrowed down to ten key-driven genes (Arc, Btg2, Dusp1, Egr3, Egr4, Fos, Ier2, Junb, Nr4a1, Npas4) and four initial genes (Cck, Egr1, Egr2, Nptx2). The "GO_Function" enrichment analysis elucidated the functional characteristics of the identified key-driven and initial genes, and showed that the functions of these DEGs were enriched in the following categories: "binding", "transcription regulator activity", "catalytic activity" and "molecular transducer activity" (Fig. 8c). We also performed "GO_biological processes" analysis and found that these DEGs were relative to "Nervous system development", "Neuron differentiation", "Brain development", "Dentate gyrus development", "Central nervous system neuron development". These findings indicate that nicotine regulates the biological processes related to delaying senesce and rescuing the functions in the aged brain ($p$ value <0.01) (Fig. 8d).

Based on the bioinformatics analysis, we carried out experiments to test nicotine-induced neurogenesis during the brain aging. We found that, in the aged brain, nicotine increased the Doublecortin (Dcx) neurons in the dentate gyrus region of the hippocampus (Fig. 8e). It is also well-known that BDNF is an important protein that

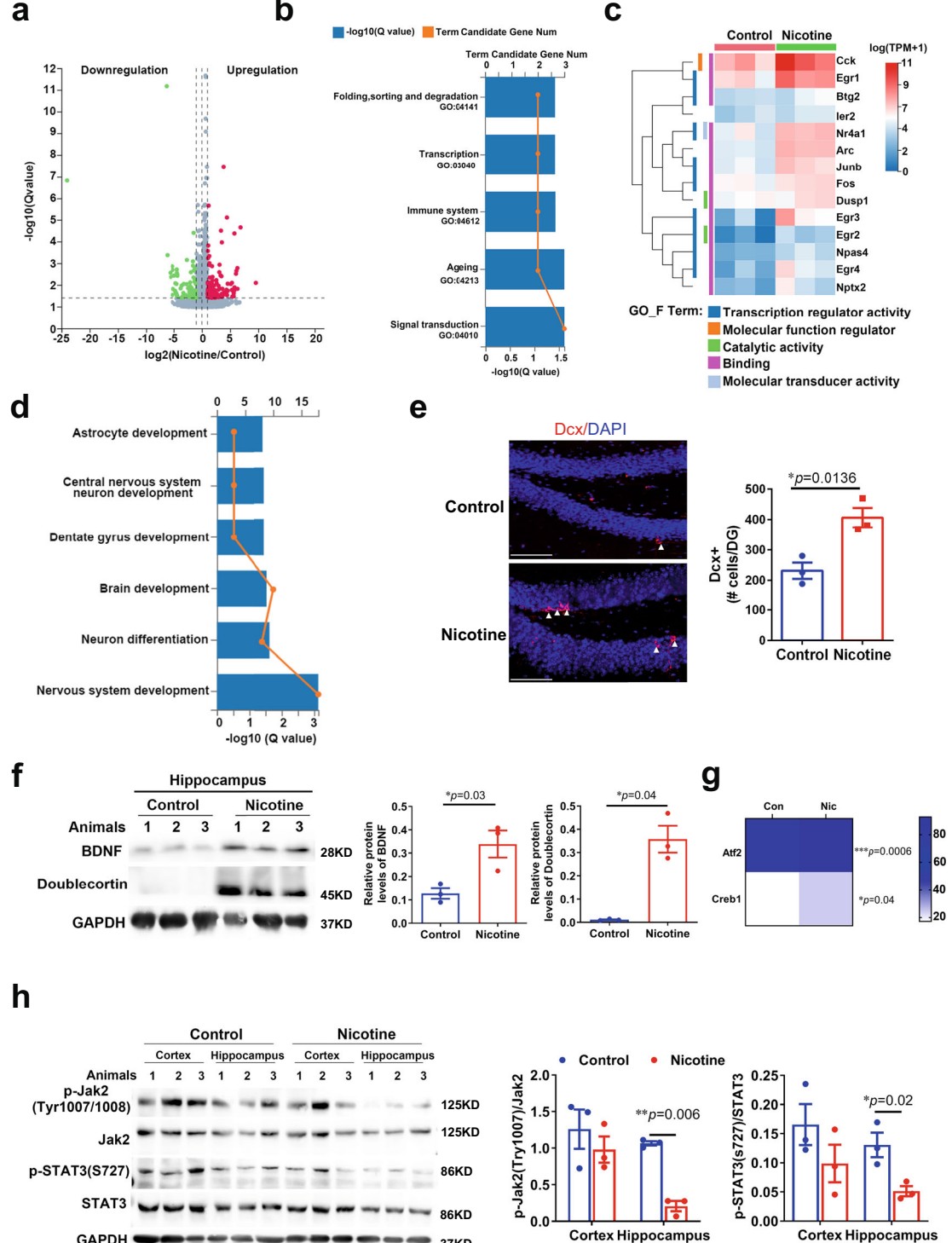

**Fig. 8 | Nicotine altered gene Expression of the aged mice brain and promoted neurogenesis and inhibited inflammation. a** Volcano plots of differentially expressed genes ($n = 3$, biologically independent samples/group). **b** The Gene KEGG pathways related to "Signal transduction", "Ageing", "Immune system", "Transcription" between control mice and nicotine mice after nicotine administrated for 6 months ($n = 3$, biologically independent samples/group). **c** Heatmap depicting transcriptional profiles of KDA gene and Go function enrichment ($n = 3$, biologically independent samples/group). **d** The "GO_biological processes" were related to neurogenesis: nervous system development, neuron differentiation, brain development, dentate gyrus development, central nervous system neuron development ($n = 3$, biologically independent samples/group). **e** Representative microscopic fields and quantification of Dcx-positive cells in the dentate gyrus (DG) of the hippocampus of naïve aged mice administered with or without nicotine ($n = 3$, biologically independent samples/group arrowheads point to individual cells; scale bar, 100 μm). Dapi, 4′, 6-diamidino-2-phenylindole. **f** Western blot and quantification of the protein expression of BDNF, Doublecortin in the hippocampus of aged mice administered with or without Nicotine ($n = 3$ biologically independent samples/group). Quantification is normalized to GAPDH. **g** Heatmap depicting transcriptional profiles of creb1 and Atf2 in aged mice brain after nicotine administration ($n = 3$, biologically independent samples/group). **h** Western blot and quantification of the protein expression of JAK2 and STAT3 in the cortex and hippocampus: phosphorylate-JAK2/JAK2 ratio and phosphorylate-STAT3/STAT3 ratio of aged mice administered with or without nicotine ($n = 3$, biologically independent samples/group). Quantification is normalized to GAPDH. Data are means ± SEM. $p$ values were determined by two-sided Student's $t$ test (**e**–**g**), or two-way ANOVA analysis and Fisher's least significant difference (**h**).

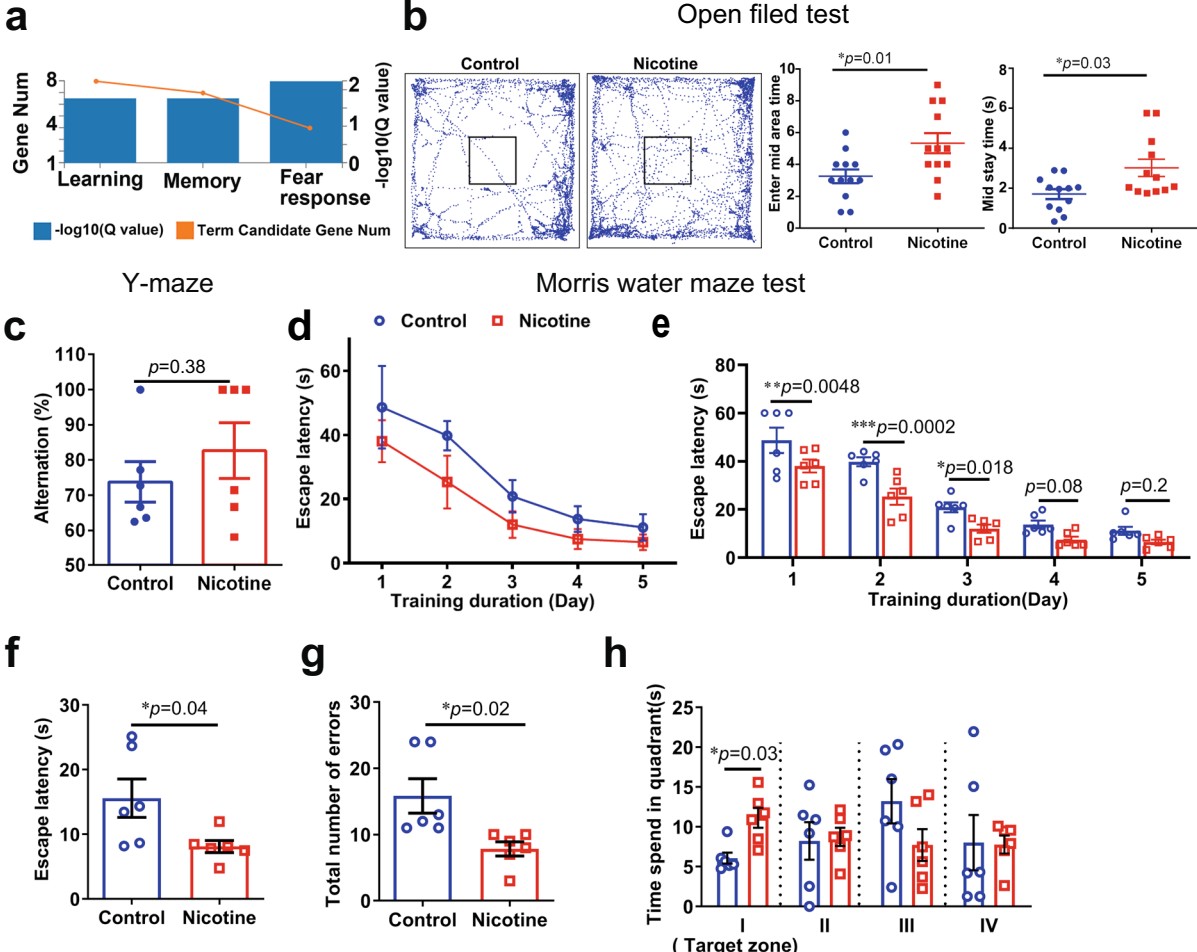

**Fig. 9 | Nicotine ameliorated cognitive function of the aged mice. a** The Gene KEGG pathways related to "behavior": learning, memory and fear response. **b** Schematic of zones in the open field arena and representative traces of with and without nicotine administration mice movement. Enter mid area times (*n* = 12, biologically independent samples/group) and mid stay time (*n* = 12, biologically independent samples/group) in open field arena. **c** The alternation of aged mice with or without nicotine in the Y-maze test (*n* = 6, biologically independent samples/group). **d–h** Behavioral performance of animals in the Morris water maze (*n* = 6, biologically independent samples/group). Average escape latencies of aged mice with or without nicotine. Escape latencies of aged mice in the explored day, total number of errors in target zone, and percentage time spent in the four quadrants. Data are means ± SEM. *p* values were determined by two-sided Student's *t* test (**b**, **c**, **f**, **g**), or two-way ANOVA analysis and Fisher's least significant difference (**e**, **h**).

influences neurogenesis and cognitive function. We examined the expression of Dcx and BDNF by western blot and observed an increased expression in the hippocampus of aged mice with nicotine treatment (Fig. 8f). We also found that nicotine significantly elevated mRNA levels of CREB1 and ATF2, which modulated BDNF transcription, and also involved in cognitive improvement and anti-inflammation[39–42] (Fig. 8g) (*p* < 0.05).

As they age, senescent cells promote the decline of NAD+ in tissue and induce inflammation[43]. The elevated pro-inflammatory state within the brain has been attributed to over-activation of microglia involved in the activation of the JAK2/STAT3 pathway[44,45]. To determine whether nicotine regulates the inflammation pathway and alters the states of inflammatory cells, we monitored phosphorylation levels of JAK2 and STAT3 in the aged cortex and hippocampus and found that nicotine significantly decreased phosphorylation of JAK2/STAT3 pathway in the hippocampus (Fig. 8h). Microglial APOE expression represented homeostatic microglial cell conversion to disease-associated microglia (DAM) and rose in response to aging[46]. In the aging brain, nicotine significantly decreased APOE mRNA expression, but other markers of DAM were not significantly changed (Supplementary Fig. 7c). Taken together, these results indicate that low-dose nicotine induces the distinct

transcriptional changes involving key-driven genes that in turn lead to neurogenesis and anti-inflammation in the aged mouse brain.

## Nicotine improved cognition of the aged mice

Supplement of NAD+ precursors improves cognitive decline[47–49]. We performed enrichment of "GO_biological processes" and found that the key-driven genes were related to "fear response", "learning" and "memory" (*p* value <0.05) (Fig. 9a). To investigate whether the long-term and nanogram-trace of nicotine administration could alter mood and cognitive functions, we performed the open field test to investigate the effect of nicotine on anxiety. In the open field test, the aged mice with nicotine spent significantly more time in the mid area and displayed less anxiety than the control mice (Fig. 9b), and the mice did not show much of fight wounds after 3 weeks of nicotine administration until the end of the study (Supplementary Fig. 9a). Next, we used Y-maze and classical Morris water maze to test the spatial memory and learning of the aged mice after 6 months of nicotine administration. Nicotine-treated mice displayed a higher ratio of spontaneous alternation than the control mice in the Y-maze test, indicating that nicotine could protect the memory of the aged mice (Fig. 9c). In the classical Morris water maze test, the escape latencies times of both control and nicotine groups gradually shortened during training days

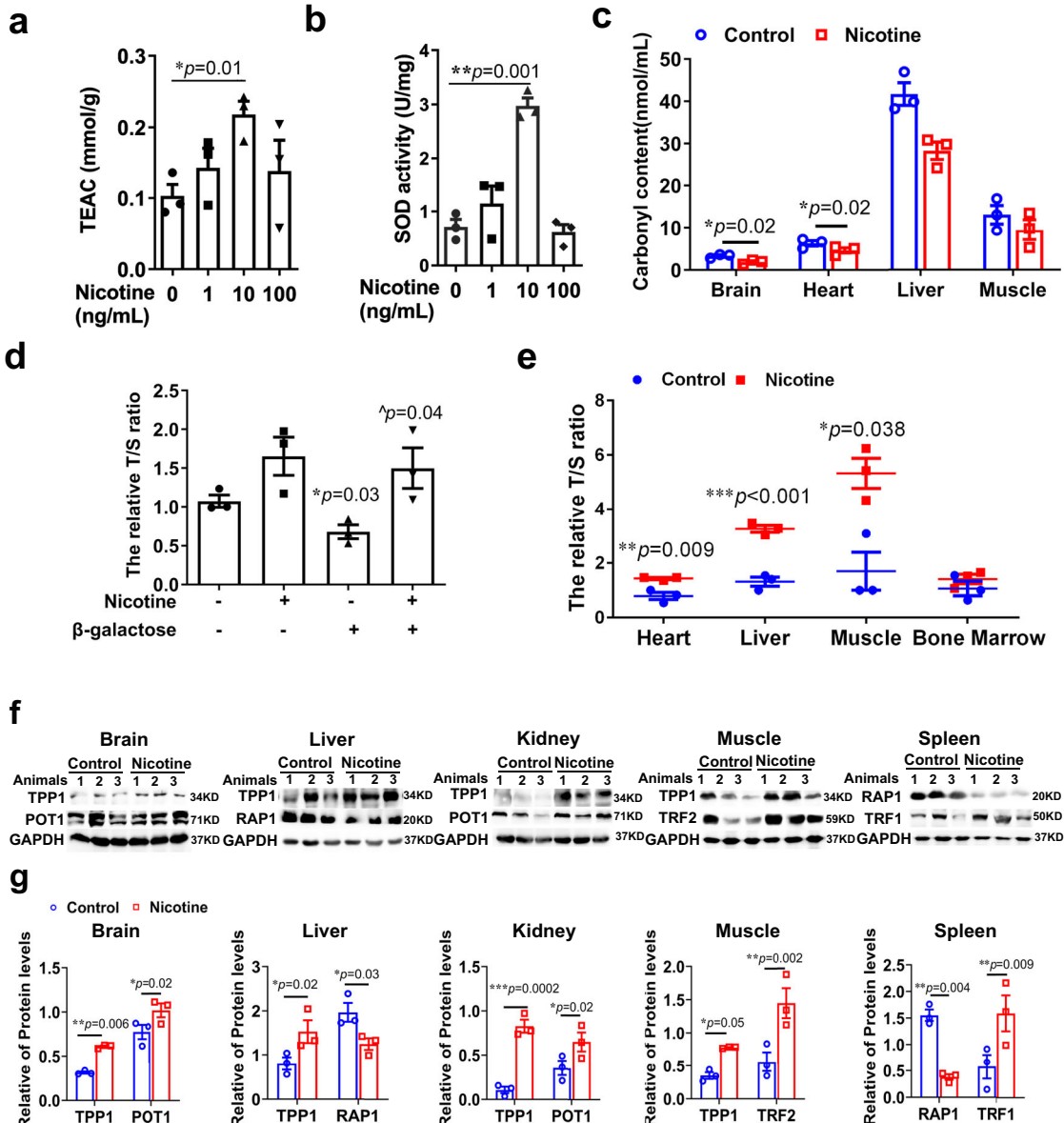

**Fig. 10 | Nicotine inhibited oxidative stress and protected Telomere stability of ageing mice. a** Nicotine 1, 10, 100 ng/mL regulated trolox-equivalent antioxidant capacity (TEAC) in HT22 cells treatment for 48 h ($n = 3$ biologically independent samples/group). **b** Nicotine 1, 10, 100 ng/mL regulated CuZn-Mn superoxide dismutase (CuZn-Mn SOD) activity in HT22 cells for 48 h ($n = 3$ biologically independent samples/group). **c** Nicotine decreased the carbonyl content in brain ($n = 3$, biologically independent samples/group), heart, liver ($n = 3$, biologically independent samples/group), and muscle of aged mice administered nicotine from 7 to 13 months. **d** Telomere length/single copy gene (T/S) ratio of β-galactose-induced ($n = 3$ biologically independent samples/group) aged HT22 cells after nicotine treatment ($n = 3$, biologically independent samples/group) for 7 days. **e** Nicotine increased the T/S ratio of heart, liver, muscle and bone marrow in aged mice administered nicotine for 6 months ($n = 3$, biologically independent samples/group). **f** Western blot and **g** were quantification of the telomere shelterin complex of brain, liver, kidney, muscle and spleen after 6 months nicotine administration. Quantification is normalized to GAPDH. Data are means ± SEM. $p$ values were determined by one-way ANOVA with Tukey's multiple comparisons tests (**a**, **b**, **d**) or two-way ANOVA analysis and Fisher's least significant difference (**c**, **e**, **g**).

(Fig. 9d), indicating that long-term administration of low-dose nicotine did not induce learning and memory impaired; while on the explore day, the nicotine mice exhibited shorter escape latencies, fewer errors, and spent more time in the quadrant containing the platform than the control mice (Fig. 9d–h). Hence, these results suggest that long-term, low-dose nicotine ameliorates the anxiety and cognitive deficits in aged mice.

## Nicotine delayed NAD⁺-decline associated systemic senesce

Given that the progression of aging is generally attributed to systemic decline[50], mechanistically, systemic decline leads to oxidative stress and telomere instability, which are affected by NAD+ depletion[51–54]. We tested the redox state of cells after administration with low-dose nicotine and found that nicotine elevated the total antioxidant capacity (TEAC) and Cu-MnSOD activity in HT22 cells (Fig. 10a, b). Protein carbonyl groups are used as the biomarkers of oxidative stress during aging[53]. We measured the carbonyl levels of the brain, heart, liver, and muscle in aged mice and found that nicotine reduced the protein carbonyl levels in these tissues (Fig. 10c). Moreover, the imbalance in NAD⁺ metabolism induces telomere shortening, and NAD⁺ precursors stabilize telomeres to slow aging[5]. To investigate whether nicotine may protect against telomere shorten during ageing, we used real-time PCR to measure the telomere length and single copy gene ratio to reflect

the relative changes in telomere length. The T/S (Telomere length/ Single copy gene) ratios of aged HT22 cells (Fig. 10d) and aged tissues, including heart, liver, muscle and bone marrow, were all significantly increased by nicotine treatment (Fig. 10e). Furthermore, telomere shelterin complex, including TRF1, TRF2, POT1, RAP1, TIN2, and TPP1, is involved in telomeres length control in cellular senescent[55]. The protein levels of TPP1, POT1, and TRF2 decrease significantly during ageing, while RAP1 increased in cell senescence and inflammatory stresses. To investigate whether nicotine regulates shelterin complex to protect the telomeres from aging, we tested the protein levels of shelterin complex in the brain, heart, liver, kidney, muscle, and spleen and found that nicotine increased the expression of TPP1, POT1, and TRF2 in most of the aged tissues. The expression of RAP1 in the liver and spleen of aged mice was significantly decreased by nicotine administration (Fig. 10f, g), but the other shelterin complex proteins were not changed (Supplementary Fig. 8a, b). Since nicotine enhances NAMPT activity levels and significantly mitigates age-related functional decline in older mice, we set up nicotine-treated and control mice to examine their survival. After 10 months of age, the nicotine-treated mice exhibited significantly lower age-related mortality than controls (Supplementary Fig. 9b). In summary, these results demonstrate that nicotine improves the oxidative stress and telomere stability in tissues.

## Discussion

As we age, NAD⁺ levels decrease, leading to a decrease in NAMPT activity, which reduces NAD⁺ synthesis and accelerates the decline of NAD⁺-dependent enzymes; this exacerbates the aging process. In this study, we demonstrated that nicotine boosts the NAD⁺ salvage pathway, restoring cognitive function and improved age-related symptoms. Long-term nanogram trace of nicotine administration increases NAD⁺ synthesis by specifically promoting SIRT1 deacetylation of NAMPT, which results in enhanced NAMPT activity, thereby promotes NAD⁺ and β-NMN biosynthesis.

Imbalanced NAD⁺ homeostasis is associated with a higher risk of cancer[56,57]. High levels of NAMPT expression and the NAD⁺/NADH ratio may contribute to tumor growth or pathological degeneration[58]. Therefore, it is not advisable to administer high doses of NAD+ precursors for long periods of time[36,59]. In contrast, nicotine could significantly increase the β-NMN levels in aged brain and liver (Supplementary Fig. 9c), and we observed that the aged mice displayed less hyperplasia and low expression of the cancer markers in the brain and liver after nicotine administration 6 months or more (Supplementary Fig. 9d).

Clinical PET imaging using the glucose analog ¹⁸F-FDG has become a standard procedure for detecting glucose metabolism characteristic of age-related disease. Previous studies have found that neuroinflammation is closely connected to this hypermetabolism, and our results from mouse brain PET images are in line with these findings[37]. Additionally, chronic systemic inflammation is a hallmark of aging, which is associated with increased glucose metabolism in the heart and other organs (Fig. 6). Nicotine has been seen to reduce systemic inflammation, partially alleviating age-related hypermetabolism, and also has systemic anti-aging effects by increasing NAD⁺ levels, which remodels glucose metabolic pathways.

NAD⁺ is a vital element of energy metabolic pathways such as glycolysis and OXPHOS[56,60]. As we age, NAMPT activity decreases and influences energetic metabolism. In hippocampal neuron cell lines, it was observed that highly acetylated NAMPT increased ATP production, potentially causing aging microglia to alter neuronal function or remove neurons[61], reducing cells' adaptability and flexibility to energy requirements during stress, which makes neurons more prone to aging and death (Fig. 7c). Furthermore, hyperacetylation levels of NAMPT significantly also affects microglia glycolysis (Fig. 7b), which could modify the microglia function and enhance the threat to senescent neurons and accelerate the aging process in the brain[62].

It has been reported that nicotine, when ingested in drinking water, is slowly absorbed through the gastrointestinal tract and is rapidly metabolized in the liver[63–66], which might explain why the concentration of nicotine in the body is extremely low (Supplementary Figs. 1a and 3c). It is well-known that long-term and high dose nicotine administration often leads to a decrease in food consumption, reduction of body weight[67,68] and diabetes-promoting actions due to Tcf7l2 highly expression in the brain. However, our study showed that the nanogram-trace nicotine decreased Tcf7l2 gene expression in the brain and did not induce increased blood sugar[69] (Supplementary Fig. 9f, g). Additionally, the body weights, the amount of food and water consumption were not changed (Supplementary Fig. 9e).

SIRT1 has been known to influences transcriptional regulation and DNA stability[20,70]. Telomere dysfunction can decreases NAD⁺ levels and suppresses sirtuin activity[54]. Conversely, SIRT1 can ameliorate senescence by modulating telomere shelterin[71]. Notably, TPP1 was the only component of shelterin that was manipulated by SIRT1 and might be the main effector of SIRT1 action. Moreover, the expression of RAP1 is impacted by SIRT1, shortening the telomere and initiating aging in the senescent cells[72,73]. Our study showed that nicotine significantly increased TPP1 and decreased RAP1 levels in most of the aged tissues to protect telomere stability via the mechanisms that might boost NAD⁺ levels and enhance SIRT1 activity (Fig. 10f, g).

In summary, nicotine possesses multiple functions in NAD⁺ homeostasis to regulate signaling pathways and metabolism involved in senescence cells. Here we show that the long-term, low-dose nicotine consumption enhances the NAD⁺ salvage pathway and delays the systemic aging, thereby, providing a supplement strategy of NAD⁺ and its precursors.

## Methods
### Animals
The animal experiments and protocols are approved by the Subcommittee on Research and Animal Care (SRAC) of Shenzhen institutes of advanced technology. C57BL/6J male mice (6 months–18 months) were housed in a specific pathogen-free facility in individually ventilated cages in an ambient temperature-and humidity-controlled room with a 12 h light/12 h dark cycle under standard housing conditions with continuous access to food and water.

### Cell culture
HT22 (BeNaCultureCollection Cat# BNCC337709) and BV2 cells (BeNaCultureCollection Cat# BNCC337749) were obtained from BeNaCultureCollection (Beijing, China) and maintained in DMEM (Sigma-Aldrich, St. Louis, MO) supplemented with 10% FBS, 100 U/mL penicillin, and 100 mg/mL streptomycin. All cells were maintained at 37 °C and 5% CO₂.

### Western blot analysis
Cells and tissues were lysed in buffer (P0013J, Beyotime), supplemented with 1:100 phosphatase inhibitor cocktail (Roche). The Protein lysis were subjected to SDS-PAGE gels, electrophoresed, transferred to to a 0.45 μm hybridization nitrocellulose Filter (HATF00010, Merck Millipore). The membrane was incubated with 5% BSA for 1 h and with primary antibodies (SIRT1 antibody; NAMPT/PBEF antibody; GAPDH antibody; ABclonal: POT1 antibody, Rap1 antibody, TPP1 antibody, TERF1 antibody, TERF2 antibody, TIN2 antibody, SIRT6, p53,PGC-1α, BDNF, Doublecortin overnight at 4 °C, then probed with horseradish peroxidase-conjugated secondary antibodies (Jackson immune research) goat anti-rabbit IgG (H + L) or anti-mouse IgG (H + L) and then developed with High-sig ECL Western blotting substrate (180-5001, Tanon) and Tanon Imaging (Tanon) were used. For quantification the Quantity One (Bio red) were used, respectively.

## Co-immunoprecipitation

Briefly, the cells and tissues were lysed in buffer (P0013J, Beyotime) supplemented with 1:100 with 1:100 phosphatase inhibitor cocktail, at 4 °C for 15 min, 12,000 × $g$ centrifuged and keep the supernatant. Protein A/G Agarose beads (sc-2003, Santa Cruz) were washed then added to samples for 30 min at 4 °C with rotation as a pre-clear step. The supernatant from samples were incubated with 10 µg antibody SIRT1 antibody; NAMPT/PBEF antibody, and the rabbit IgG (Pierce) or mouse IgG isotype control (Santa Cruz) at 4 °C overnight, and added the protein A/G agarose beads for 1 h at room temperature, then washed beads with lysis buffer for three times and beads were eluted in SDS-PAGE loading dye with DDT at room temperature for western blot analysis.

## Small-animal PET

In total, 2, 6, 16 and 16 months Nicotine mice were followed an established standardized protocol for Radiochemistry, acquisition, and post processing[74,75]. All PET experiments were performed with isoflurane anesthesia during imaging. The images were reconstructed by using ordered-subset expectation maximization (OSEM) with 16 subsets and 5 iterations. Target–to–reference tissue SUV ratios (SUVRs) were calculated for 18F-FDG[37]. The PET estimates deriving from a standardized target volume of interest (VOI; brain 40 mm$^3$ Heart 40 mm$^3$ body 360 mm$^3$) was used for scaling of 18F-FDG data. The statistical results of SUVs in the regin of interesting (ROI) area were calculated and analyzed by Amide 1.0.4-1(San Diego, CA 92101).

## Metabolic extracellular flux analysis

The bioenergetic properties of primary microglia under different conditions were determined using the XF-96 Seahorse extracellular flux analyzer (Seahorse Bioscience, CA). It measures the real-time changes in extracellular acidification rate (ECAR) and the oxygen consumption rate (OCR) that is indicative of glycolysis and mitochondrial respiration, respectively. In duplicate or triplicate of HT22, BV2 cells were plated on XF-96 cell culture plates in six groups: Control group, Nicotine group, NAMPT-Flag group, NAMPT-Flag + Nicotine group, NAMPT-K53Q group and NAMPT-K53Q + Nicotine group. These cells were washed and analyzed in the XF Running buffer A (XF assay medium, 10 mM glucose, 1 mM pyruvate sodium, 2 mM L-Glutamine) for OCR measurements or XF Running buffer B (XF assay medium, 2 mM L-Glutamine) for ECAR measurements. Measurements were obtained in real-time with no drug treatment (basal conditions) and with the sequential treatment of different drugs: 1 µM oligomycin, 2 µM FCCP, 0.5 µM rotenone/antimycin A for MitoStress assay, or 10 mM glucose, 1 µM oligomycin and 50 mM 2-DG for Glycolysis stress assay. Upon completion of the measurements, we used BCA method for protein quantification, and used for normalized analysis of each well. OCR measurement under basal conditions in the absence of drugs represents the basal OCR, and ECAR measurements after the addition of glucose represent the basal ECAR. The key parameters reflecting OCR are: Basal respiration, ATP-linked respiration, Proton leak, Maximal respiration, Spare respiratory capacity, Non-mitochondrial oxygen consumption. The key parameters reflecting ECAR: Glycolysis, Glycolytic reserve, Glycolytic capacity and non-glycolytic acidification.

## Scratch wound analysis

HT22, BV2 cells were seeded in tissue culture plates at 5 × 10$^4$ cells per well, and after 24 h of adherence, cells were transfection with NAMPT-Flag or NAMPT-K53Q for 24 h and treated with nicotine. All the cells were using IncuCyte Wound Maker to make wounds and the detached cells were removed using PBS wash. We observed cell migration in real time by IncuCyteS3, took pictures every 2 h, and then following the Analysis Guidelines of Incucyte Scratch Wound Analysis Software Module.

## The NAMPT-Flag and NAMPT-K53Q mutation overexpression

The plasmids of NAMPT-Flag and NAMPT-K53Q mutation were transfected with Lipofectamine 3000 according to the manufacturer's instructions (Invitrogen) to HT22, BV2 cells in triplicates. After 12 h, these cells were treatment with nicotine (10 ng/mL) for 48 h. These cells were measured by western blotting, metabolic extracellular flux analysis or scratch wound analysis. The plasmids of targeting mouse NAMPT-Flag and NAMPT-K53Q were purchased from GenePharma.

## RNA interference

The sirt1, sirt6, siRNAs were transfected with Lipofectamine 3000 according to the manufacturer's instructions (Invitrogen) to HT22 cells in triplicates. After 12 h, HT22 cells were treatment with nicotine (10 ng/mL) for 48 h. HT22 cells were measured by western blotting or NAD$^+$-NADH assay. The siRNA oligonucleotides targeting mouse sirt1 and sirt6 were purchased from GenePharma.

## T/S ratio

The Average telomere length was measured from total genomic mouse or HT22 cells DNA (Genomic DNA Mini Preparation Kit (Spin Column), #D0063, Beyotime) by using a real-time quantitative PCR method previously described[76,77]. The amount of the PCR product approximately doubles in each cycle of the PCR, the T/S ratio is approximately $[2Ct (telomeres)/2Ct (36B4)]-1 = 2^{-\Delta Ct}$. The relative T/S ratio is $2^{-(\Delta Ct1-\Delta Ct2)} = 2^{-\Delta\Delta Ct}$.

## Carbonyl content

The tissues carbonyl protein content was determined according to instructions provided with the protein carbonyl colorimetric assay kit (no. 10005020) purchased from Caymanchem.

## NAD$^+$ detection

NAD$^+$ levels were determined according to instructions provided with the NAD/NADH assay kit purchased from Abcam.

## NAMPT activity

The acute NAMPT activity was determined according to instructions provided with the Cyclex NAMPT Colorimetric assay kit purchased from Medical & biological laboratories. CO. LTD.

## Nuclear SIRT activity

The nuclear SIRT activity was determined according to instructions provided with the Universal SIRT Activity Assay Kit (Colorimetric) purchased from Abcam®.

## LC-MS

In total, 1 × 10$^7$ cells or 0.5 mg accurately weighed sample was transferred to a 1.5 mL Eppendorf tube, and Liquid nitrogen quenching for 30 min. In total, 0.5 mL mixture of methanol and water (1/1, vol/vol) were added to each sample, samples were placed at −80 °C for 2 min. Then grinded at 60 HZ for 1 min, and ultrasonicated at 4 °C for 2 min. Samples were centrifuged at 13,000 × $g$, 4 °C for 15 min. In total, 0.3 mL of supernatant were collected using crystal syringes, filtered through 0.22 µm microfilters and transferred to LC vials. The external standard method was selected to carry out the quantitative analysis. Good linearity with all of the correlation coefficients ($R^2$) = 0.9999 was acquired, as calculated with the peak area of selected SRM transitions (Y) to the concentration of analyte (X, ng/ml)[78–82].

## Total antioxidant capacity and CuZn/Mn-SOD activity

The total antioxidant capacity and CuZn/Mn-SOD activity was determined according to instructions provided with the Total Antioxidant Capacity Assay Kit and Cu/Zn-SOD and Mn-SOD Assay Kit purchased from Beyotime.

## NAD⁺/NADH biosensor state

HT22 cells were transfected with NAD⁺-NADH redox states fluorescent biosensor[83]: pcDNA3.1-Peredox-mCherry (Addgene: #32383) and pcDNA3.1-Peredox-mCherry-NLS (Addgene: #32384). The plasmids were transfected with Lipofectamine 3000 according to the manufacturer's instructions (Invitrogen) to HT22 cells in triplicates. After 12 h, HT22 cells were treatment with nicotine (10 ng/mL) for 24 h. HT22 cells were imaged 48 h after transient transfection. Green fluorescence was obtained with 488 nm excitation and red fluorescence was obtained with 575 nm. Images were acquired using an Olympus inverted fluorescence microscope. The mean fluorescence intensity was quantified by using ImageJ, and generated a green-to-red ratio for NAD⁺/NADH ratio.

## Intercellular Ca2⁺

The plasmid of $Ca^{2+}$ indicator: (Addgene: #50942) pAAV-hSyn1-mRuby2-GSG-P2A-GCaMP6s-WPRE-pA were transfected to HT22 cells. The plasmids were transfected with Lipofectamine 3000 according to the manufacturer's instructions (Invitrogen) to HT22 cells in triplicates. After 12 h, added the working concentration (1–1000 ng) nicotine near to cells. The fluorescence was obtained with 488 nm excitation and Images were acquired using an Olympus inverted fluorescence microscope. The mean fluorescence intensity was quantified by using ImageJ.

## Immunofluorescent staining

Mice were deeply anesthetized with 1% pentobarbital sodium (50 mg/kg) and transcardially perfused with PBS followed by 4% PFA. Mice brain was removed, and fixed in 4% PFA overnight at 4 °C and cry protected in 30% sucrose at 4 °C for 48 h. Immunohistochemistry experiments were performed on free-floating coronal 30 μm cryostat slices. The following antibodies were used 1:200 rabbit Doublecortin (abcam ab207175). The HT22 cells were perfused with ice-cold PBS and incubated in 4% PFA for 1 h and 1% Triton X-100 for 10 min, 5% bovine serum albumen (BSA) for 1 h then incubated overnight at 4 °C with the following primary antibodies: SIRT1 D1D7 (Cell Signaling Technology #9475s), NAMPT /PBEF (Santa Cruz sc-393444). All the primary antibodies were visualized with secondary antibodies conjugated with Alexa fluorophores (Invitrogen) and counterstained with DAPI.

## Open filed test

The apparatus was a gray PVC-enclosed arena 50 × 9 × 30 cm, divided into a 10 cm squares. Mice were placed in the center of a novel open field environment. A mouse was placed into a corner square facing the corner and observed for 5 min. Mice were allowed to explore the arena for 10 min, and time spent, distance traveled, and entries into the center (36 cm² area) of the arena were recorded. The movement of the mouse around the field was recorded with a video tracking device for the entire testing period.

## Y-maze

The functional behavior of 12-months-old age male mice littermates from each group (Control, Nicotine). The Y-maze apparatus consisted of three arms (A, B, and C) (40 cm × 8 cm × 15 cm) with an angle of 120° to each other and connected by a central zone (CZ). All mice were placed in one of the terminal identical arms to freely explore for 5 min without any added stress such as lights, sound, and food deprivation. In a series of explorations, the body completely entered the arm for standard entry. The total numbers of entering the each arm (n) were recorded. The mice entering three different arms in succession were considered as the correct alternating reactions. Spontaneous alternation rate (%) = [correct number of alternation reactions/(n − 2)] × 100.

## Morris water maze test

The Morris water maze assay was conducted as described in ref. [84]. The Morris water maze consisted of a circular pool (diameter, 1.2 m; depth, 55 cm) filled with water (21 ± 1.5 °C) and painted white. The water pool was divided into four quadrants (I, II,III, and IV). A round platform (10 cm in diameter) was submerged 1.0 cm below the water surface in the center of quadrant I (target quadrant). The test involved the spatial probe trial and a 1-day probe trial. During the spatial probe trial, the mouse was placed randomly into the pool facing the wall individually from four different preset starting points and allowed to swim. The escape latency to find the platform and the related parameters were recorded. If the mouse reached the platform and stayed for 10 s within 60 s, the test would end automatically. If the mouse could not find the platform, it would be guided to the platform by the experimenter and allowed to stay on it for 10 s, with the escape latency recorded as 60 s. One day after the last training session, the probe trial was conducted in which the platform was removed and the mice were allowed to swim freely for 60 s. The data were collected and analyzed by Ethovision XT.

## RNA sequencing of aged mice brain

In this project, we sequence four samples used BGISEQ platform, averagely generating about 6.48 GB bases per sample. The average mapping ratio with reference genome is 91.75%; 18,050 genes were identified. Sequencing Platform: BGISEQ, Sequencing length: PE150The raw data contains reads of low quality, reads with adaptor sequences and reads with high levels of N base. Those reads need to be filtered before the data analysis for reliability analysis results. We used HISAT to align the clean reads to the reference genome. We used Bowtie2 to align the clean reads to the reference genes. Differentially expressed genes (DEGs) were called using edgeR for each experimental design. Functional profiling of DEGs was performed using g: profiler and Gene Ontology and KEGG databases and Key-driven gene analysis of BGI Bioinformatics analysis platform.

## Mice blood sugar test

The C57Bl/6J mice mice were fasting for about 4 h, and their blood glucose was measured between 14:00 and 15:00 in the afternoon. The mice tail vein blood were measured by glucometers.

## Statistical analysis

Most of the experiments were repeated at least three times and the exact $n$ is stated in the corresponding figure legend. All other statistical analysis was performed with GraphPad Prism 8.0 software. Multiple groups were tested using analysis of variance (ANOVA) with one-way ANOVA with Tukey's multiple comparisons test and comparisons between two groups were performed using two-tailed two-way ANOVA with Fisher's LSD. The data are shown as the mean ± SEM. Differences were considered statistically significant when $p < 0.05$. Asterisks indicate levels of significance (*$p \leq 0.05$; **$p \leq 0.01$; ***$p \leq 0.001$).

## Reporting summary

Further information on research design is available in the Nature Portfolio Reporting Summary linked to this article.

## Data availability

All data supporting the findings described in this manuscript are available in the article and in the Supplementary Information and from the corresponding author upon request. The RNA sequencing data of mouse brain is available in the NCBI SRA BioProject database, BioProject ID: PRJNA910147. Source data are provided with this paper.

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

## Acknowledgements

This work was supported by grants from Shenzhen Science and Technology Program (KQTD20210811090117032), Shenzhen Key Laboratory of Viral Vectors for Biomedicine (ZDSYS20200811142401005), CAS Key Laboratory of Brain Connectome and Manipulation (2019DP173024) and Guangdong Provincial Key Laboratory of Brain Connectome and Behavior (2017B030301017).

## Author contributions

L.Y. supervised, conceived, and coordinated the studies, wrote the paper, performed and analyzed the experiments in all figures. N.L. and J.S. performed LC-MS experiments; M.G. performed Seahorse analyzer; C.L., Y.T. and Y.Z. performed Morris water maze and Y-maze

experiments; Y.Y., Z.K. and Z.Q. performed PET experiments. X.L. supervised, conceived, and coordinated the studies, and wrote the paper. All authors reviewed the results and approved the final version of the manuscript.

## Competing interests

The authors declare no competing interests.
