## [Peer Review File · Nature Communications]

Nicotine rebalances NAD⁺ homeostasis and improves aging-related symptoms by enhancing NAMPT activityREVIEWER COMMENTS

Reviewer #1 (Remarks to the Author):

It has been reported that NAMPT, the rate-limiting enzyme in the salvage pathway of NAD biosynthesis has the anti-aging effect, which is mediated by increasing NAD content and sirt1 activity. In this manuscript, the authors demonstrated that low-dose nicotine can reverse the age-related decline of NAMPT activity by enhancing SIRT1 binding to NAMPT and promote deacetylation of NAMPT, which in turn boosts NAD⁺ synthesis, as well as rebalance energy metabolism and biological function. In general, this topic is interesting for a broad readership, dealing with roles and the molecular mechanisms of low-dose nicotine delaying the aging. However, there are still some concerns as noted below:

1. There are several major discrepancies on results described in the text vs. those shown in Figures. For example, the text in Results stated that “we observed that nicotine could reverse the dysfunction of glycolytic and mitochondrial respiration in BV2 cells (Fig. 6A, B) and slightly improve mitochondrial respiration in HT22 (Fig. 6C).” However, the results can't be found in Fig. 6A-C.
2. The aged mice used in the study were mostly 12 months old. However, in the part of glucose hypermetabolism (Fig.6), 16-month-old mice was used.
3. The sample size of this study in RNA-seq was too small.
4. Ten key-driven genes (Arc, Btg2, Dusp1, Egr3, Egr4, Fos, Irf2, Junb, Nr4a1, Npas4) and 4 initial genes (Cck, Egr1, Egr2, Nptx2) were obtained through RNA-seq. However, further research on these genes is lacking.
5. In the present study, nicotine can promote neurogenesis and inhibited inflammation in aged mice. Whether nicotine can regulate BDNF expression and JAK2/STAT3 pathway in BV2 microglia?
6. Although English language is readable, it still requires some fine tuning.

Reviewer #2 (Remarks to the Author):

This is an extensive study by Drs. Yang and Li looking at several different aspects on how nicotine may be of benefit in countering age-related detrimental outcomes. The concept is noteworthy as any intervention that may delay the aging process might be a welcome news. Moreover, the results may add to plethora of other postulated beneficial effects of nicotine.

I have, however, few reservations on recommending publication in Nature Communications.

1. First and foremost, the paper is wide and in my opinion is more suited for a Journal that specifically deals with aging process.
2. There are 2 major concerns regarding their conclusion that nicotine may actually delay the aging process. One is that they never actually looked at the life-span to prove this point. Second, the association of NAD⁺ homeostasis with other numerous factors involved in aging process, demonstrating it to be both a necessary and sufficient factor for the eventual delay of the aging process, is lacking.
3. The authors claim that the action of nicotine is independent of nicotinic receptors because they did not observe changes in gene expression of nAChRs subunits alpha and beta, or in Ca⁺⁺ entry into HT22 cells. There could be shortcoming with both reasonings as, in general, nicotinic receptor function may be independent of the gene expression or Ca⁺⁺ entry. They could have simply used a nicotinic receptor antagonist such as mecamylamine or more selective antagonists such as dihydro- β -erythroidine (DH β E) or Methyllycaconitine to rule out involvement of nAChRs.

There are also numerous typos and grammatical errors in the manuscript, few of which are indicated below.

Line 36. “was efficient for anti-aging”

Line 47. "was also coordinately regulate"
Line 48. Sentence is incomplete
Line 50. "In deed"
Line 70. Space missing before (
Line 75. Use of "interacted" instead of "interaction"
Line 82. "to enhancing"
Line 126. Space missing before "of"
Line 181. Should read "nAChRs" and not "nAchRs"

Response to reviews

Reviewer #1 (Remarks to the Author):

It has been reported that NAMPT, the rate-limiting enzyme in the salvage pathway of NAD biosynthesis has the anti-aging effect, which is mediated by increasing NAD content and sirt1 activity. In this manuscript, the authors demonstrated that low-dose nicotine can reverse the age-related decline of NAMPT activity by enhancing SIRT1 binding to NAMPT and promote deacetylation of NAMPT, which in turn boosts NAD⁺ synthesis, as well as rebalance energy metabolism and biological function. In general, this topic is interesting for a broad readership, dealing with roles and the molecular mechanisms of low-dose nicotine delaying the aging. However, there are still some concerns as noted below:

1. There are several major discrepancies on results described in the text vs. those shown in Figures. For example, the text in Results stated that “we observed that nicotine could reverse the dysfunction of glycolytic and mitochondrial respiration in BV2 cells (Fig. 6A, B) and slightly improve mitochondrial respiration in HT22 (Fig. 6C).” However, the results can’t be found in Fig. 6A-C.

Answer: Thanks for the careful reviewing.

We apologize for the figure numbering error. The related description is now presented in Figure 7 and in result section (line 19, page 8) of the revised manuscript. We have corrected the figure number: “Notably, we observed that nicotine could reverse the dysfunction of glycolytic and mitochondrial respiration in BV2 cells (Fig. 7A, B) and slightly improve mitochondrial respiration in HT22 (Fig. 7C).”

2. The aged mice used in the study were mostly 12 months old. However, in the part of glucose hypermetabolism (Fig.6), 16-month-old mice was used.

Answer: Thanks for the careful reviewing.

In general, the senescence phenotype is already present in C57bl/6j mice at 12 months of age, so we used 12-month-old mice for most of our experiments. We also examined glucose metabolism in 12-month-old mice using FDG-PET (see below).To further

determine the beneficial effects of nicotine on aging, we used 16-month-old mice to demonstrate nicotine-treated effects.

3. The sample size of this study in RNA-seq was too small.

Answer: Thanks for the comment.

We added the number of sequences in each group. The related description is now presented in Fig.8(A, C and G) and Fig.S7A,B in result section (line1, page 27) of the revised manuscript. The RNA sequence transcriptomic data is available in the NCBI SRA BioProject database. BioProject ID PRJNA910147, <http://www.ncbi.nlm.nih.gov/bioproject/910147>.

4. Ten key-driven genes (Arc, Btg2, Dusp1, Egr3, Egr4, Fos, Ier2, Junb, Nr4a1, Npas4) and 4 initial genes (Cck, Egr1, Egr2, Nptx2) were obtained through RNA-seq. However, further research on these genes is lacking.

Answer: Thank you very much for your comments, which will be very helpful for our future research.

These genes are involved in many signaling pathways and a variety of biological

processes. The immediate-early genes IEG (egr-1, c-fos, Arc, Npas4) and Egr4 have important roles in processes such as brain development¹, learning and memory formation². Nptx2 is a component in the regulation of anxiety and memory³. Egr2 regulates myelination of the peripheral nervous system^{4, 5}. Egr3 regulates the neurobiological processes such as BDNF, synaptic plasticity, memory and cognition⁶. Although we did not study the effects of nicotine on the functions of these genes in depth, we examined the functions related to these genes in this paper, such as learning and memory by Y maze and water maze, emotion by open field test, and neurogenesis, brain BDNF levels, and brain inflammation levels by immunofluorescence and WB experiments. These experiments implicated that nicotine affects neurogenesis and synaptic plasticity, which are closely related the biological functions of these key-driven genes.

5. In the present study, nicotine can promote neurogenesis and inhibited inflammation in aged mice. Whether nicotine can regulate BDNF expression and JAK2/STAT3 pathway in BV2 microglia?

Answer: Thanks for the comment.

As an immortalized microglia, BV2 cell retained the function and morphology of the primary microglia. LPS (Lipopolysaccharide) stimulation could activate BV2 cells, and we found that nicotine inhibits the activation of the JAK2/STAT3 inflammatory signaling pathway.

Because BV2 cells do not express the BDNF gene, we demonstrated that nicotine increases BDNF levels in the hippocampus of the brain (Figure 8F).

6. Although English language is readable, it still requires some fine tuning.

Thanks the comments, we have made some improvements of the language in the article. We will work with the editor to further refine the language.

Reviewer #2 (Remarks to the Author):

This is an extensive study by Drs. Yang and Li looking at several different aspects on how nicotine may be of benefit in countering age-related detrimental outcomes. The concept is noteworthy as any intervention that may delay the aging process might be a welcome news. Moreover, the results may add to plethora of other postulated beneficial effects of nicotine.

I have, however, few reservations on recommending publication in Nature Communications.

1. First and foremost, the paper is wide and in my opinion is more suited for a Journal that specifically deals with aging process.

Answer: Thanks for the advice.

This article is an initial exploration of the overall effects of low-dose nicotine on aging mice, involving changes in various organs and the role of various molecular mechanisms in aging. It is comprehensive research, so we think it is appropriate to submit it in Nature Communication.

2. There are 2 major concerns regarding their conclusion that nicotine may actually delay the aging process. One is that they never actually looked at the life-span to prove this point. Second, the association of NAD⁺ homeostasis with other numerous factors involved in aging process, demonstrating it to be both a necessary and sufficient factor for the eventual delay of the aging process, is lacking.

Answer: Thanks for the comments.

1. According to our experimental records and literature reports, the anti-aging effect of NAD⁺ did not change the longest life span of mice, while the main effect of nicotine is to improve aging symptoms. We now add to these results on the effects of

nicotine on lifespan in mice, demonstrating that supplementation with nicotine in middle-aged to elderly mice does not extend lifespan. However, the nicotine-treated elderly mice generally maintained a much healthier appearance and higher activity than the control mice. We presented the related description in Fig.S9B and in result section (line327, page12) of the revised manuscript.

2. The cellular NAD⁺ levels decline during chronological aging. This decline appears to play a crucial role in metabolic dysfunction and the development of age-related diseases. A large number of studies have reported that NAD⁺ homeostasis affects aging⁷⁻⁹ including inflammation, energy metabolism, mitochondrial function, REDOX, and genomic stability. We found that nicotine treatment could improve brain inflammation and reduce oxidative stress in aging animals by increasing NAD⁺ (Fig.8H); nicotine also increased the total antioxidant capacity and SOD2 activity (Fig. 10 A,B) and decreased the carbonyl protein levels in tissues (Fig. 10D); telomere length and telomere complex protein levels in tissues were protected (Fig.10 E,F).

3.The authors claim that the action of nicotine is independent of nicotinic receptors because they did not observe changes in gene expression of nAChRs subunits alpha and beta, or in Ca⁺⁺ entry into HT22 cells. There could be shortcoming with both reasonings as, in general, nicotinic receptor function may be independent of the gene expression or Ca⁺⁺ entry. They could have simply used a nicotinic receptor antagonist such as mecamylamine or more selective antagonists such as dihydro-β-erythroidine (DHβE) or Methyllycaconitine to rule out involvement of nAChRs.

Answer: Thanks for the comment.

Based on the suggestion, we used nAChRs inhibitors to rule out that nicotine

regulates NAD⁺ metabolism through activating nicotinic receptors. We found that nAChRs inhibitor D-Tubocurarine pentahydrate does not affect the effect of nicotine on SIRT1 and NAMPT binding, nor does it affect the effect of nicotine on NAD⁺ generation. The related description is now presented in Fig.S1G and in result section (line187, page7) of the revised manuscript.

There are also numerous typos and grammatical errors in the manuscript, few of which are indicated below.

Line 36. “was efficient for anti-aging”

Line 47. “was also coordinately regulate”

Line 48. Sentence is incomplete

Line 50. “In deed”

Line 70. Space missing before (

Line 75. Use of “interacted” instead of “interaction”

Line 82. “to enhancing”

Line 126. Space missing before “of”

Line 181. Should read “nAChRs” and not “nAchRs”

Answer: Thanks for the comment. We corrected the grammar and wording errors.

References:

- [1] Bae, C.-J., Jeong, J., and Saint-Jeannet, J.-P. (2015) A novel function for Egr4 in posterior hindbrain development, *Scientific reports* 5, 7750.
- [2] Minatohara, K., Akiyoshi, M., and Okuno, H. (2015) Role of Immediate-Early Genes in Synaptic Plasticity and Neuronal Ensembles Underlying the Memory Trace, *Frontiers in molecular neuroscience* 8, 78.
- [3] Chang, S., Bok, P., Tsai, C.-Y., Sun, C.-P., Liu, H., Deussing, J. M., and Huang, G.-J. (2018) NPTX2 is a key component in the regulation of anxiety, *Neuropsychopharmacology* 43, 1943-1953.
- [4] Niu, R. N., Shang, X. P., and Teng, J. F. (2018) Overexpression of Egr2 and Egr4 protects rat brains

- against ischemic stroke by downregulating JNK signaling pathway, *Biochimie* 149, 62-70.
- [5] Wüst, H. M., Wegener, A., Fröb, F., Hartwig, A. C., Wegwitz, F., Kari, V., Schimmel, M., Tamm, E. R., Johnsen, S. A., Wegner, M., and Sock, E. (2020) Egr2-guided histone H2B monoubiquitination is required for peripheral nervous system myelination, *Nucleic Acids Res* 48, 8959-8976.
- [6] Pfaffenseller, B., Kapczinski, F., Gallitano, A. L., and Klamt, F. (2018) EGR3 Immediate Early Gene and the Brain-Derived Neurotrophic Factor in Bipolar Disorder, *Frontiers in behavioral neuroscience* 12, 15.
- [7] Covarrubias, A. J., Perrone, R., Grozio, A., and Verdin, E. (2021) NAD⁺ metabolism and its roles in cellular processes during ageing, *Nature Reviews Molecular Cell Biology* 22, 119-141.
- [8] Imai, S.-i. (2016) The NAD World 2.0: the importance of the inter-tissue communication mediated by NAMPT/NAD⁺/SIRT1 in mammalian aging and longevity control, *npj Systems Biology and Applications* 2, 16018.
- [9] Imai, S.-i., and Guarente, L. (2016) It takes two to tango: NAD⁺ and sirtuins in aging/longevity control, *npj Aging and Mechanisms of Disease* 2, 16017.

REVIEWERS' COMMENTS

Reviewer #1 (Remarks to the Author):

Thanks for the response. My concerns have been addressed by the authors in this revision.

Reviewer #2 (Remarks to the Author):

The authors have done a good job in accommodating my concerns. However, I have one important correction that has to be incorporated in the title of the paper as well as in the text. As it stands, one would conclude from the title that nicotine actually increases the life span by delaying aging. The authors now provide evidence that this is not the case and that in reality only the age-related symptoms are improved. Thus, in the title as well as in the abstract and throughout the text, the term "delay in aging" has to be avoided. I would suggest changing the title to: "Nicotine rebalances NAD⁺ homeostasis and improves aging-related symptoms by enhancing NAMPT activity" and in the abstract and throughout the text emphasize improvement in age-related symptoms. Please avoid the use of any of the following: "anti-aging", "delay aging", "preventing aging" etc.

There are still some typos and/or syntax corrections that I assume can be addressed by the publisher. For example, should use "nAChR" not "nACHR"